# Phage Therapy: Towards a Successful Clinical Trial

**DOI:** 10.3390/antibiotics9110827

**Published:** 2020-11-19

**Authors:** Andrzej Górski, Jan Borysowski, Ryszard Międzybrodzki

**Affiliations:** 1Bacteriophage Laboratory, Hirszfeld Institute of Immunology and Experimental Therapy, Polish Academy of Sciences (HIIET PAS), 53-114 Wroclaw, Poland; ryszard.miedzybrodzki@hirszfeld.pl; 2Phage Therapy Unit, Hirszfeld Institute of Immunology and Experimental Therapy, Polish Academy of Sciences (HIIET PAS), 53-114 Wroclaw, Poland; 3Infant Jesus Hospital, The Medical University of Warsaw, 02-005 Warsaw, Poland; 4Department of Clinical Immunology, Transplantation Institute, Medical University of Warsaw, 02-006 Warsaw, Poland; jborysowski@interia.pl

**Keywords:** phage therapy, clinical trial, antibiotic resistance

## Abstract

While phage therapy carried out as compassionate use (experimental therapy) has recently flourished, providing numerous case reports of supposedly healed patients, clinical trials aiming to formally prove their value in accord with current regulatory requirements have failed. In light of the current issue of increasing antibiotic resistance, the need for a final say regarding the place of phage therapy in modern medicine is evident. We analyze the possible factors that may favor success or lead to the failure of phage therapy: quality of phage preparations, their titer and dosage, as well as external factors that could also contribute to the outcome of phage therapy. Hopefully, better control of these factors may eventually bring about long-awaited positive results.

## 1. Introduction

Interest in phage therapy has rapidly grown, which is reflected by the greatly increased number of case reports describing patients being treated. Furthermore, a number of journals have recently launched special issues and research topics dedicated to phage therapy (Front Microbiol., Front Pharmacol., Viruses, Antibiotics, Pathogens, Microorganisms, etc.). What is more, reviews on phage therapy appear almost every month. However, this “phage therapy fever” has not been accompanied by any reports that could have confirmed the true value of phage therapy in accord with the current standards of evidence-based medicine. In other words, no double blind randomized clinical trial has so far yielded data supporting the promising observations derived from experimental therapy in animals and humans. The most recent failure was reported by Leitner et al. [1] who were unable to demonstrate the superiority of phage therapy over a placebo or antibiotics in patients with urinary tract infections to whom phages were administered intravesically. Reflecting on this failure, Pirnay and Kutter reviewed other trials that also failed to provide positive data: (1) the Phagoburn trials examining the efficacy and tolerability of a phage cocktail in the treatment of burn wound infections with *Pseudomonas aeruginosa*, and (2) acute coliform diarrhea in children treated with coliphages (in the latter case, failure was associated with a paucity of target bacteria in the guts of children subjected to phage therapy with coliphages). The authors also mention the first controlled trial of phage therapy performed in 1966 based on *Staphylococcus aureus* lysate injections to treat infections in asthmatic children that also yielded negative results [2]. Table 1 presents relevant points of critical clinical trials completed thus far. Evidently, we need a critical analysis of completed clinical trials to identify possible causes of their inability to confirm the therapeutic value of clinical phage therapy. Such analysis would help to identify the most relevant factors contributing to the success or failure of a clinical trial. The short commentary of Pirnay and Kutter mentions some of these factors while the aim of the present communication is to discuss this dilemma in a more systematic way [2].

## 2. Quality and Titer of Phage Preparations

An important issue that may determine the anti-bacterial activity of phages is their titer and the quality of the administered phage preparations. Worley–Morse et al. demonstrated a linear relationship between bacterial inhibition and phage concentrations for some phages, while others were active even at a lower titer (10^5^/mL) [5]. In a Phagoburn trial, a preparation with a relatively low phage titer was initially applied (10^6^/mL). Subsequently, the titer decreased and the need to dilute the preparation to a lower endotoxin concentration reduced the daily phage dose to 10–100 PFU/mL instead of the expected concentration of 1 × 10^6^/mL [4]. The Pyophage preparation used in the trial of Leitner at al. is not well characterized, as it “is composed of multiple individual phages active against a range of bacteria with the titre of 10^4^/mL for streptococcus phages and 10^5^/mL for other phages” [1]. Notably, only in Phagoburn trial phage the preparations have been produced following GMP practices [4]. The phage titers used in those two recent, highly important trials were rather low and their biological activity in vivo could be questioned. In contrast, a recent summary of activities performed at the Phage Therapy Center in California indicates that phages in a titer of 10^10^/mL administered intravenously (iv) can produce beneficial effects without any significant side effects but such side effects can appear when the titer is increased to 10^11^/mL [6]. In addition, the same phage concentration has been successfully used iv in the treatment of prosthetic knee infection [7] and good clinical results have been achieved using a 10^9^/mL phage concentration in the treatment of ventilator-associated pneumonia and empyema [8].

Since phage pharmacokinetics is at an early stage of determination, little is known about the optimal dosage and therapeutic phage concentrations in patients’ sera. In this regard, Schooley et al. achieved good results when this concentration reached 10^4^/mL [9]. Petrovic-Fabijan reported similar results in patients with serum levels of 2 × 10^5^ PFU/mL, which was attained using a total dose of 10^9^ administered phages [10]. Dedrick et al. reported serum concentrations as high as 10^9^/mL and higher when a dose of 10^9^ phages was administered every 12 h. [11]. It seems that serum monitoring for phage concentrations may be helpful in assuring optimal trial conditions, at least in trials where systemic phage administration is required.

## 3. Phages and Antibiotics

Another relevant issue that may determine the success of phage therapy is concomitant use of antibiotics. Data has been accumulating that supports the synergistic anti-bacterial effects of such a phage–antibiotic combination [12,13,14,15,16]. It is therefore noteworthy that in all the clinical trials completed so far, phages were used as a stand-alone therapy. In contrast, antibiotics have frequently been used with phages in successful compassionate use approaches [6,7,10,17]. At the present time it is impossible to exclude that such combinations could be an optimal approach in the treatment of difficult bacterial infections and therefore a clinical trial on phage therapy might also include a subgroup of patients treated with both phages and antibiotics. The results of such a study could finally supply an answer to the dilemma of whether antibiotics with phages are indeed superior to phage therapy alone. However, this strategy of combined treatment with phages and antibiotics may not be recommended for all phages and antibiotics: recent data indicate that aminoglycoside antibiotics inhibit Mycobacteriophage DNA replication and therefore interfere with pathogen elimination by phages [18].

## 4. Phage Therapy as Precision Medicine

When reflecting on possible causes of the mentioned failures, especially the trial carried out by Leitner at al., the authors highlight the value of a personalized therapy concept where phages are matched for their activity against bacteria isolated from the infected patients. That approach allows the invading pathogen to be precisely targeted and has been extensively used in compassionate use phage therapy [1]. However, the current manufacturing realities, pharmacoeconomic models and marketing requirements favor predefined phage cocktails that have been used in phage therapy clinical trials so far. In addition, preliminary data might suggest that phage cocktails are more immunogenic than monovalent phage preparations, which could adversely affect the outcome of their use [19]. Apparently, the dilemma of a personalized phage approach vs. phage cocktails needs more studies.

We believe that the use of ‘personalized’ phage preparations is in line with the principles of precision medicine. Precision medicine has been defined as “The tailoring of medical treatment to the individual characteristics of each patient…to classify individuals into subpopulations that differ in their susceptibility to a particular disease or their response to a specific treatment. Preventative or therapeutic interventions can then be concentrated on those who will benefit, sparing expense and side effects for those who will not” [20]. Thus, the most important feature of precision medicine is that it enables doctors to use the most effective treatment for a given patient while reducing the burdens associated with unnecessary and potentially ineffective diagnostic tests and therapies [21]. The principles of precision medicine are also considered in the context of treatment of bacterial infections [22,23]. The fundamental feature of bacteriophages—that is, their capacity to eliminate target pathogenic bacteria without adversely affecting the microbiota—seems to be perfectly in accord with the philosophy of precision medicine.

Personalized phage therapy, which consists of matching a phage to a bacterial isolate, requires a collection of phages usually referred to as phage bank. As pointed out by Yerushalmy et al. [24], a phage bank should contain well-characterized phages that are ready to be used in clinical trials against as many strains of bacteria as possible. A well-known source of phages for researchers is ATCC. Furthermore, the Phage Directory [25] is meant to enable communications related to the search for therapeutic phages and their possible exchange between phage laboratories and phage collections. Further expansion of phage banks and the development of their networks could guarantee optimal exchange and distribution of phages for clinical trials and phage therapy similar to international organ exchange and organizations responsible for those activities (Eurotransplant at a pan-European level and national exchange organizations, e.g., Poltransplant for Poland, United Network for Organ Sharing (UNOS) for USA, etc.).

## 5. Phage Interactions with Eukaryotic Cells May Modify the Effects of Phage Therapy

Recent studies indicate that phages may interact with human cells and such interactions may enhance phage anti-bacterial action. For example, Shan et al. showed that phages can reduce bacterial numbers much more effectively in the presence of human cells. This process is phage and cell specific—which is similar to phages being able to interact only with their target bacteria. The phenomenon probably depends on specific phage receptors for definite human cells although the exact mechanism is unclear at the present time [26]. If this phenomenon is also relevant for in vivo phage interactions with bacteria and human cells then it could also affect the outcome of phage therapy: phage effects could be potentiated by cells of a given organ or tissue being able to enhance their anti-bacterial action in contrast to cells of a different body location without this phage enhancing ability. However, at the present time, there are no available data on such effects regarding cells from, for example, the human urinary tract or lungs.

Further studies on phage interactions with cells derived from different organs and tissues may suggest a clinical setting in which clinical trials would have high chances of success.

## 6. Antibody Responses during Phage Therapy

More than half a century ago, Jerne established that phages are immunogenic and elicit antibody production that can inhibit their anti-bacterial activity [27]. However, it is rather surprising to note that such responses have received little or no attention at all in cases when the outcome of phage therapy has been analyzed [28]. At the same time, it has been recognized that the effectiveness of phage therapy will also depend on the immune response to the administered phage [29]. In most cases, the formation of such antibodies has not been monitored even when patients received phages intravenously (iv) [6,10,16,29,30]. Notably, Dedrick et al. noted only very weak antibody responses and no evidence of phage neutralizing antibodies even though the patients were on iv phage therapy for 32 weeks [11]. Our group has studied the issue of antibody responses during phage therapy in some detail showing that they depend on phage type and titer, the route of administration (very weak following oral route) and the length of phage therapy. Furthermore, there appears to be no clear association between the level of antibody responses and the outcome of phage therapy [19]. However, very limited data are available on antibody levels and the outcome of phage therapy in patients receiving iv phages. Therefore, further clinical trials should also include antibody monitoring to clarify this dilemma.

## 7. External Factors

It is well known that phages vary in their sensitivity to external factors [31]. For example, tea infusions have been shown to inactivate some phages [32,33]. Furthermore, phages vary in their susceptibility to alcohol, which may abolish the activity of T-even but not T-odd phages [34]. An insightful implementation in a clinical trial should also include a standard diet (including drinks!) and concurrent medications. In trials involving oral phage administration one has to keep in mind that drugs neutralizing gastrin acid may affect phage activity, e.g., *S. aureus* A5/80 phage is sensitive to aluminum-containing preparations, while ranitidine (a H2 receptor antagonist) decreased the titer of both T4 and A5/80 phages to 2%. Interestingly, yoghurt was 6x more effective in neutralizing acid and enabling phage transit to the small intestine. In this regard, the effect of yoghurt (but not milk) was comparable to that caused by a proton pump inhibitor, omeprazole [35].

Our recommendation for future trials:Use of a well-characterized phage preparation (GMP standards recommended);Phage preparation should have appropriate phage titer (preferably > 10^6^/mL and dosage (at least 10^9^/patient);Use of personalized phage preparations rather than predefined phage cocktails;Monitoring for serum anti-phage antibodies (especially during iv therapy).

## 8. Conclusions

In the face of the current issue of increasing antimicrobial resistance we urgently need to determine whether phage therapy offers a reliable tool to successfully confront this challenge. To confirm its medical value a clinical trial showing the positive effects of the therapy is of decisive value. In this communication we point to a few factors that could contribute to a breakthrough in clinical trials of phage therapy.

## Figures and Tables

**Table 1 antibiotics-09-00827-t001:** Summary of most important phage therapy trials.

Year	Target	Results	Factors Responsible for Failure	Reference
2016	*E. coli*/*Proteus* (children diarrhea)	Lack of clinical efficacy of oral phages	Insufficient phage coverage; too low *E. coli* titers; overgrowth of *Streptococcus*	[3]
2017	*P. aeruginosa*/*E.coli* (burn infections)	Trial was stopped because of insufficient efficacy	Decreased titer of phage preparations	[4]
2020	*Staphylococcus* *Streptococcus* *E. coli* *P. aeruginosa* *Proteus* (urinary tract infections)	Success rates similar to placebo (bladder irrigation)	Reduction of the bacterial load caused by bladder irrigation with placebo comparable with the effect of phage preparations	[1]

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
