# Peer review of "Phage Therapy: Towards a Successful Clinical Trial"

_antibiotics, 2020, doi:10.3390/antibiotics9110827_

Round 1

Reviewer 1 Report

The article is concise, well written and deals with the factors explaining the success or failure of phage therapy. However, its clarity should be improved with some modifications. First, the aims and content of the manuscript should be stated early on. There is no introductory paragraph where the general idea of the work is stated. We do not know what to expect or the approach. Then, the structure of the manuscript may be improved. The topics included in the article are :

1) Most important phage therapy clinical trials

2) Personalised or “precision” Medicine

3) Phage banks

4) Titer and quality of phage preparations

5) Combining phage therapy with antibiotics

6) Positive interactions of phages with human cells

7) Immune response to phages

8) Diet factors influencing efficacy of phages

A more structured flow could be organized as follows:

a) General considerations of phage therapy clinical trials: 1, 4, 5

b) Changing the model of clinical trials, from pre-manufactured phage cocktails to personalized preparations: 2,3

c) Further factors to be considered: 6,7,8

In the first section, the reasons of failure of previous clinical trials are not organized or clearly explained. A chronological order may help too, as trials were done with a wide time lapse. A small table with the clinical trials so far would be very useful: year, target, results, factors of failure involved, etc. The clinical trial on children suffering bacterial diarrhea is not well referenced. Please include where necessary :

Oral Phage Therapy of Acute Bacterial Diarrhea With Two Coliphage Preparations: A Randomized Trial in Children From Bangladesh.

Sarker SA, Sultana S, Reuteler G, Moine D, Descombes P, Charton F, Bourdin G, McCallin S, Ngom-Bru C, Neville T, Akter M, Huq S, Qadri F, Talukdar K, Kassam M, Delley M, Loiseau C, Deng Y, El Aidy S, Berger B, Brüssow H. EBioMedicine. 2016 Jan 5;4:124-37. doi: 10.1016/j.ebiom.2015.12.023. eCollection 2016 Feb.

Abstract : no abbreviations (PT), please. In the main text, some abbreviations are confusing, such as in vivo and intravenously (iv both). No need to abbreviate in all cases.

Citation : who is sir William Osler?

51-52 . Reference to that statement

Author Response

An introductory paragraph was added.

The structure of the manuscript was re-oragnized according to the instructions given.

A table entitled ‘Summary of most important phage therapy trials’ was included.

A trial on children with diarrhea was included (Ref. 3 by Sarker et al.)

All abbreviations ‘PT’ were replaced with term ‘phage therapy’.

To avoid confusion this citation is deleted.

Reviewer 2 Report

The review describes the factors that may favor success or lead to the failure of phage therapy. The covered content is comprehensive with appropriate and recent citations.

Although I recommend the publication of the manuscript, I suggest to the Authors some minor revisions:

  • The work should be better organized in its three points, as reported in the abstract (page 1, lines 19-20).
  • Phage preparations methods should be described and compare with methods used at this data, highlighting because the GMP is the recommended standard method, as reported in page 4 line 153.

Author Response

The manuscript was re-organized (see response to Ref 1).

The GMP status of the phage preparations was specified (line 63).

Reviewer 3 Report

The current manuscript describes factors influencing success of clinical phage therapy.

It would be better read if the authors can compartmentalize the contents according to the four points they made- 1. Use of a well-characterized phage preparation, 2. phage preparation should have appropriate phage titer, 3. Use of personalized phage preparations rather than predefined phage cocktails, 4. Monitoring for serum anti-phage antibodies. Also, various clinical cases can be seen handily if they are organized in a Table.

Minor points

1) line 32, ͈”PT fever“

2) line 44, ͈”infective asthma“

3) line 80, 105/ml --> 105 PFU/ml

4) line 81, 106/ml --> 109 PFU/ml

5) line 84, ͈“is composed ~~

6) line 86, 105/ml --> 105 PFU/ml

7) line 89, 1010/ml --> 1010 PFU/ml

8) line 91, 1011/ml --> 1011 PFU/ml

9) line 96, 104/ml --> 104 PFU/ml

10) line 98, 109/ml --> 109 PFU/ml

11) line 104, [18,19,20,21,22] --> [18-22]

Author Response

The content of the manuscript has been changed (see response to Ref 1).

Minor points corrected (5: a citation from the original work is used).
